# A Survey of 6D Object Detection Based on 3D Models for Industrial Applications

**DOI:** 10.3390/jimaging8030053

**Published:** 2022-02-24

**Authors:** Felix Gorschlüter, Pavel Rojtberg, Thomas Pöllabauer

**Affiliations:** 1Fraunhofer-Institut für Graphische Datenverarbeitung, Fraunhoferstraße 5, 64283 Darmstadt, Germany; pavel.rojtberg@igd.fraunhofer.de (P.R.); thomas.poellabauer@igd.fraunhofer.de (T.P.); 2Department Graphisch-Interaktive Systeme, Technische Universität Darmstadt, Karolinenplatz 5, 64289 Darmstadt, Germany

**Keywords:** object detection, pose estimation, machine learning, neural networks, synthetic training, RGBD

## Abstract

Six-dimensional object detection of rigid objects is a problem especially relevant for quality control and robotic manipulation in industrial contexts. This work is a survey of the state of the art of 6D object detection with these use cases in mind, specifically focusing on algorithms trained only with 3D models or renderings thereof. Our first contribution is a listing of requirements typically encountered in industrial applications. The second contribution is a collection of quantitative evaluation results for several different 6D object detection methods trained with synthetic data and the comparison and analysis thereof. We identify the top methods for individual requirements that industrial applications have for object detectors, but find that a lack of comparable data prevents large-scale comparison over multiple aspects.

## 1. Introduction

The problem of 6D object detection comprises the detection of objects and the estimation of the translation and rotation thereof. In a three-dimensional space, both of these properties have three degrees of freedom, thus resulting in the 6D portion of the term. In many cases, algorithms that solve this problem also give an estimate of the target object’s class (in this work, the term *object detection* implies *object classification*). The most common sensors used here to record scenes are cameras. In this work, we focus on approaches for solving this problem with two specific properties:RGBD cameras (i.e., color and depth) are available for providing input to the algorithm;Only 3D object models (CAD or reconstructed) are required to set up the algorithm (i.e., no recordings by real cameras).

Algorithms with these properties are especially well suited for industrial applications, specifically automation tasks. On the one hand, RGBD images are easy to obtain in production environments. We have mostly indoor scenes with controlled lighting, simplifying the usage of active sensors. The larger form factor, compared to RGB cameras, is usually no problem in static setups, and the pricing of high-quality RGBD sensors does not weigh heavily on company-scale budgets. On the other hand, industrially manufactured products are usually based on computer-aided design (CAD), which makes the 3D models of target objects readily available. There are two major use cases in industrial environments that require localizing real-world objects: robotic manipulation and quality control, examples of these are shown in Figure 1.

In this work, we examine the current state of the art of 6D object detection for application in industrial use cases. We put a strong focus on empirical data; to our knowledge, we collected the most comprehensive comparison of evaluation scores for object detectors with the aforementioned properties to date. Our core contributions are the following:A listing of requirements that typical industrial use cases have for object detectors.A comprehensive collection of empirical data from experiments with 6D object detectors that meet the identified criteria.Empirical data on the performance of the object detector FFB6D [2], which has not been evaluated with purely model-based training, yet.

In the remainder of this work, we first give an overview of related work. Then, we present the background of our work by presenting a definition of the 6D object detection task, establishing a rationale for our focus by identifying the requirements of typical industrial applications and giving a short overview of model-based training (strictly speaking, the term *training* refers to setting up learning-based algorithms. For better readability, in this work, we also use it to refer to generating reference data for non-learning-based algorithms) and synthetic data generation for this purpose. We then describe the method of our analysis, including a categorization of examined algorithms and a description of used datasets and metrics. Finally, the collected data are presented and discussed, followed by a short conclusion.

## 2. Related Work

In this section, we give an overview of publications relevant to 6D object detection for industrial applications, starting with listing reviews and benchmarks in this area, then presenting individual object detectors, their specific contributions and finally the state of the art of model-based training and the generation of synthetic data for the training of object detectors.

### 2.1. Reviews and Benchmarks

Hodaň et al. [3] performed a large-scale benchmark of 6D object detectors in their *BOP Challenge 2020*. They provided seven datasets known from the literature in a uniform format and added synthetic images for each one, generated with *BlenderProc* [4], a set of scripts allowing physically based renderings of procedurally generated scenes with *Blender* (https://www.blender.org/, accessed on 30 December 2021). They tested 26 different methods and found that CosyPose [5] had the best overall score, as well as the best score for synthetically trained object detectors, under their metric. In addition to their paper, they published their evaluation results on the accompanying website (https://bop.felk.cvut.cz/home/, accessed on 30 December 2021), which is still extended with new evaluation data and new datasets.

The most comprehensive review to date regarding 6D object detection was published in 2020 by Sahin et al. [6]. They presented a categorization of respective methods based on their mathematical models, compared empirical results thereof and identified challenges of different datasets.

The specific use case of robotic manipulation was addressed in 2021 by Cong et al. [7] and Du et al. [8]. Like Sahin et al. [6], they categorized different object detectors and datasets. They also collected a large amount of quantitative data to compare methods. However, unlike prior reviews, they also considered metrics and datasets specific to grasp estimation.

We used the references above as a starting point for our research, but extended the data discussed in these works with the evaluation results for algorithms not yet available at the respective time of writing. We also put the data in a new context by putting the focus of our research on the requirements of industrial use cases as defined in Section 3.2).

### 2.2. 6D Object Detectors and Pose Estimators

Here, we list different approaches to 6D pose estimation and the novelties presented in their respective publications. Bold keywords mark methods that fit the requirement profile that set the scope for this work and thus were considered in our analysis.

In 2010, Drost et al. [9] presented an approach for detecting objects with known 3D models in point clouds (or depth images). They relied on calculating features based on distances and normal angles between two object points, called *point pair features (****PPFs****)*. Hinterstoisser et al. [10] improved PPFs by introducing more robust sampling and voting schemes, which were developed even further by Vidal et al. [11]. Their approach made first place in the *BOP Challenge 2018* [12].

In 2011, Hinterstoisser et al. [13] performed 6D object detection by matching templates against input images, calling their method **LineMOD**. The matching is done in a feature space describing RGB- and depth gradients. They improved their method by refining the strategy to sample poses for training and by introducing a filtering strategy based on object color [14]. LineMOD was further extended by Rios-Cabrera and Tuytelaars [15] to **DTT-OPT-3D** by discriminatively learning templates with SVMs. In 2014 and 2018, Tejani et al. integrated LineMOD-features into a patch-based regression forest, calling the resulting algorithm **latent-class Hough forest (LCHF)** [16,17]. Another **template-based** approach was presented by Hodan et al. [18] in 2015. Unlike LineMOD, theirs is based on voting schemes.

In 2014, Brachmann et al. [19] presented an approach based on random forests, predicting 2D–3D correspondences, from which poses were estimated using RANSAC. This work was extended to auto-context random forests in [20] and labeled ***uncertainty-driven***
*pose estimation*. In 2016, Kehl et al. [21] published a regression- and voting-based method, using a **convolutional autoencoder (CAE)** on RGBD images. The same authors presented an extension of the *single shot pose (SSD)* algorithm to **SSD6D** in 2017 that works on RGB. In 2017, Buch et al. [22] presented an object detector based on subgroup voting and **pose clustering** that uses constraints posed by oriented points of two models. Rambach et al. [23] explicitly tackled the task of learning **object poses from synthetic images** in 2018. They tried to bridge the domain gap by letting their network operate on edge-filtered images. Tekin et al. [24] published **YOLO6D** in 2018, whose key contribution was letting a CNN predict projections of the 3D bounding box corners of objects and using the gained 2D–3D correspondences to solve for the pose using the PnP algorithm. In 2018, Sundermeyer et al. [25] presented **augmented autoencoders (AAEs)**, which build upon denoising autoencoders, and addressed the synthetic-to-real domain gap by training their autoencoder in a way that makes it invariant to the gap. Park et al. [26] published **Pix2Pose**, predicting the 3D coordinate of an object per pixel and reconstructing the pose using a RANSAC-based PnP algorithm. **Dense pose object detector (DPOD)** by Zakharov et al. [27] works in a similar way, but also employs an RGB-based refinement. Thalhammer et al. [28] presented **SyDPose**, which again explicitly addresses using only synthetic data for training. The **coordinate-based disentagled pose network (CDPN)** by Li et al. [29] predicts translation and rotation, separately. **PointVoteNet** by Hagelskjar and Buch [30], unlike most other neural-network-based methods, estimates poses from unordered point clouds. **CosyPose** by Labbé et al. [5] also supports multi-view pose estimation and was one of the top performers in the *BOP Challenge 2020* [3]. **EPOS** by Hodaň et al. [31] represents objects as compact surface fragments. **SynPo-Net** by Su et al. [32] converts training- and input images into an edge-filtered representation before prediction to bridge the domain gap. **PoseRBPF** by Deng et al. [33] considers rotation and translation separately, using a Rao–Blackwellized particle filtering framework. He et al. [34] presented PVN3D, a network relying on keypoint detection and Hough voting, building on the DenseFusion features by Wang et al. [35]. They expanded this work in 2020 with bidirectional fusion of RGB- and depth-features, calling the resulting method **FFB6D** [2]. **SurfEmb** by Haugaard and Buch [36] introduces a contrastive loss.

### 2.3. Model-Based Training and Image Synthesis

In this section, we present works that address the task of object detection when only a 3D model of the target object is available at the training time. A particular challenge here is rendering (“synthesizing”) images that are suitable for training learning-based detectors.

Rudorfer et al. [37] found that rendering on random backgrounds outperforms rendering in a realistic context via rigid body simulation with a static background. In 2019, Denninger et al. [4] presented *BlenderProc*, a Blender-based rendering pipeline that creates synthetic images based on physically based rendering (PBR). They provided realistic lighting and different modalities, such as normal maps and depth images. This was later used in BOP Challenge 2020 to provide synthetic training images. Hodan et al. [38] compared physically based rendering with realistic lighting, surfaces, object placement and scene context for rendering random photographs and found that physically based rendering outperforms the latter approach. This finding was confirmed by Hodaň et al. [3] in the context of their BOP Challenge 2020. For the problem of 2D object detection, Hinterstoisser et al. [39] showed that detectors based on synthetic training can outperform detectors trained with real images under the right circumstances. They specifically focused on applying domain randomization to the renderings and creating images with good viewpoint coverage. Rojtberg et al. [40] utilized GANs to learn the difference between real and synthetic images and then transform synthetic images into the real domain based on these networks. They found that this strategy cannot reach the performance of real images, but increases performance compared to pure domain randomization. Eversberg and Lambrecht [41] investigated the effect of different strategies for reducing the domain gap between real and synthetic images, focusing on object detection only. They found that image-based lighting using high dynamic range images and using random real images as backgrounds is beneficial for synthetic training; they recommended using at least 5000 images. Rambach et al. [23] and Su et al. [32] tackled the domain gap by first applying various augmentations and then bringing both synthetic training images as well as real input images into a common *pencil filter domain*. They found that this strategy increases the accuracy of synthetically trained object detectors.

## 3. Background

This section provides background information on concepts referenced in this work and sets the scope for our analysis. Specifically, we define the problem of *6D object detection*, describe requirements that industrial applications pose, describe what constitutes *model-based training* and give an overview of the modalities we took into account in this work.

### 3.1. Problem Definition: 6D Object Detection

*Six-dimensional object detection* comprises the detection of objects and an estimation of their three-dimensional translation and their three-dimensional rotation. We define the relationship of *6D object detection*, *object detection* and *6D pose estimation* as follows:6Dobjectdetection=objectdetection+6Dposeestimation.
i.e., a *6D object detector* detects object instances in a scene and outputs their locations as 6D poses. For some detection methods, this is a single algorithmic step (often called *single-stage detectors*, e.g., [9,26,34]), while others perform object detection and pose estimation as distinct steps (*two- or multi-stage detectors*, e.g., [42,43,44]). The latter usually first employs an object detector that outputs 2D bounding boxes for object instances found in an image and then inputs these into a pose estimator.

Hodaň et al. [3] differentiated *object detection* and *object localization* in Appendix 1 of their work. When *detecting* objects, one tries to find an unknown number of objects, while *localizing* objects means that we know a priori that *N* objects are visible in the scene and we need to find their locations. In our specific case, *localizing* an object means estimating its 6D pose. Thus, in this work, the term *object localization* is synonymous to *6D pose estimation for N objects*, which, according to the relationship given above, makes *object localization* a sub-task of *6D object detection*.

The common usage of *object detection* in the literature also implies the classification of objects. In this work, we assume 3D models of objects as references to look for, i.e., we look at object detectors that perform their task based on very specific geometric properties, which is known as *instance-level detection*. This stands in contrast to *category-level detection*, whose goal is to detect objects that fall into broader categories, for instance, “find all cars in an image”.

Similar to Sahin et al. [6], we define 6D pose estimation formally as
(1)$$T_{i}^{*}=\underset{T_{i}}{arg max}P\left(T_{i}|I,S,O\right),$$
where $T=(r_{1},r_{2},r_{3},t_{1},t_{2},t_{3})$ is the six-dimensional pose of object instance *i*, *I* is the input image, *S* is a seen instance of an object, and *O* the reference for an object class. Pose estimators try to maximize the probability function *P*. In practice, different pose estimators mainly alternate in their formulation of *P*, e.g., some use neural networks [27,34,45], while others use hand-crafted heuristics to determine the probability [9,14]. The output of *P* can also be interpreted as the *detection score*.

Whether a use case is a detection or localization task has two important practical implications:The parametrization of the algorithm is different. For localization, we can accept the *N* best hypotheses that the object detector produced, while for detection, we need to set a score threshold for *P* as an acceptance criterion for the hypotheses.The required metrics for evaluating the performance differ. For localization, determining a score that only regards the rate of positive detection is sufficient (e.g., recall). As the detector outputs a maximum of *N* results, we know that every false positive implies a false negative, e.g., the precision is always at least as good as the recall here. For detection tasks, this is not true, and so we need to regard metrics that both take true and false positives into account (e.g., recall and precision).

### 3.2. Industrial Applications

There is a great potential in object detectors when applied to problems that occur in industrial environments. Particular tasks from the areas of quality control and robotic manipulation require fast and accurate detection and pose estimation of target objects. To deduce the requirements that industrial use cases have for object detectors, we identified their chances and challenges regarding 6D object detection. Chances are aspects of these scenarios that potentially simplify 6D object detection, while challenges are those that make it harder. Chances are as follows:CAD models are available, which means that generating reference data is cheap.High-end and RGBD cameras are available, as higher costs and a larger form factor compared to RGB cameras are negligible in large-scale production environments.Scene setups are controlled. Production mostly happens indoors, and the placement of lights and cameras can be controlled easily. Indoor setups also allow for a broader range of possible RGBD cameras, as active cameras often do not work well in sunlight.The minimally required frame rate for many automation tasks is the production’s takt time, which is usually lower than the required frame rate for interactive applications.

We conclude that the combination of RGBD cameras and 3D models as references makes optimal use of these chances. RGBD allows for a higher robustness and accuracy than RGB, and the availability of 3D models allows generating synthetical images, which can be acquired much more simply and cheaply than annotated real-world recordings. On the other hand, we find that industrial applications pose the following specific challenges for 6D object detection:Lots of industrially manufactured objects are **textureless**. Specifically, workpieces that are at the beginning of production chains are often made of a single material with flat and untextured surfaces.A lot of man-made objects, especially those with simple geometry, are **rotationally symmetric**, or at least appear so under certain perspectives. This makes their poses ambiguous, which can be a difficult problem for algorithms relying on optimization.A common task in the area of robotic manipulation is bin picking. Here, individual objects can be highly **occluded**.Additionally, especially in bin-picking tasks, we have an **unknown number of instances** of the same object class. As described in Section 3.1, we refer to this task as object detection in contrast to object localization, where the number of objects to detect is known a priori. When attempting to detect an unknown number of instances, false positives can be a major problem.Object **colors are often unspecified** in the reference data. CAD models generally store an object’s geometric and kinetic properties, but not its surface properties, defining color and reflective behavior.There are objects with **difficult surface properties** that hinder the recognition of geometric properties based on optical recordings, i.e., objects made from materials with high specular reflections, such as metals, or objects made from translucent or transparent materials, such as glass.

Here, too, RGBD can alleviate the problems posed by these challenges. The geometric information encoded in the depth channel can complement the color information and lead to better accuracy when there is no discerning texture on the target object or no color information in the reference. Especially in use cases that target a lot of object instances, much can be gained by generating annotated synthetic scenes, showing heaps of objects, as manual annotation is hardly feasible for those.

Of course, not necessarily all of these characteristics apply to every industrial use case. However, based on our experience, we find that these properties are typical for production environments and so they set the scope of this work. In the remainder, we analyze if and how good published object detectors can fulfill the requirements posed by the presented chances and challenges.

### 3.3. Model-Based Training

In this section, we give a definition of what constitutes model-based training and give an overview of how synthetic images can be generated. For object detectors that only regard the geometric properties of target objects, model-based training is straightforward. This kind of algorithm can be trained directly with the reference model by generating features in a latent space, e.g., PPFs [9] fall in this category. Training object detectors that work on surface properties and projections of model geometries (i.e., on images) is more involved. This is especially true for learning-based algorithms, which generally work better the more similar the data available at training time are to the input data during inference. Generating “real images”, i.e., recordings of target objects for estimation and annotating them with ground-truth poses, is a very involved and costly process. This is especially true if the use case requires training images taken under a multitude of different perspectives, lighting conditions or from different objects. Synthetic images, on the other hand, which comprise *simulated* recordings, are cheap, and images taken under a great number of different simulated conditions can be generated easily. To achieve this, a 3D model of the target object has to be available, which is the case in industrial production environments, as most products are usually modeled before they are manufactured. From these models, one can derive rendered images that also take into account the properties of the recording process. We found the following strategies of utilizing models of the real world in order to train object detectors:**3D models:** Here, we directly derive features in the latent space from the information contained in a 3D model, i.e., a model’s vertices and normals, e.g., PPFs only require a 3D model of an object at training time.**Augmented real images:** In this strategy, real images are augmented to generate a higher variety of training images. This can be done by simulating varied recording conditions, e.g., changing an image’s size or aspect ratio, its brightness or sharpness, or adding noise. A more involved mode of image augmentation is the “Render and Paste” strategy in which an object is cropped from its original scene and pasted onto a different background to simulate a varying background, or covered by another cropping to simulate occlusion.**Renderings:** Rendering is the process of simulating the full image recording pipeline and thus generating 2D images from 3D models. There is a big variety in how this simulation is implemented and how realistic the resulting output is. The simplest and quickest method for rendering images is using a rasterization-based renderer, such as OpenGL. This type of renderer usually produces plausible, but not necessarily physically accurate renderings, in order to achieve real-time performance. A more ambitious mode of generating realistic images is called *physically based rendering (PBR)*, which is not a strictly defined term, but usually entails more realistic simulation of the behavior of light and surfaces than the commonly used Blinn–Phong model [46], e.g., by employing ray tracing.

In this work, the term *model-based trained object detectors* refers to algorithms that are trained either on 3D models only or renderings thereof (synthetic images), i.e., the training of these algorithms does not include recording the physical target objects. Note, however, that we do not exclude algorithms that use training images involving generic real-world images as backgrounds, textures or distractors, as these images can be easily obtained from 2D image datasets, such as ImageNet (https://www.image-net.org/, accessed on 23 December 2021).

We did not investigate the nature of each training set used to train the methods referenced here. This means that it is likely that some methods could perform considerably better by training them on images generated with more advanced strategies for synthetic image generation. Therefore, the values presented should be regarded as the empirically proven *lower bound* for each algorithm’s performance.

### 3.4. Modalities

As described in Section 3.2, we assume the availability of a RGBD camera for industrial use cases and include 6D object detectors that take RGBD images as input. For a potential user, the quality of the output of a method is much more relevant than which modality it accepts, as long as it is compatible with available hardware. As RGB-based detectors are fully compatible with RGBD images, we also include the former in our survey. Note that RGB-based detection can easily be refined with depth information by employing a geometry-based refinement algorithm, such as ICP [47].

There are two modalities that could fit the requirements of industrial use cases as defined in Section 3.1 well, but they were *not* explicitly considered in this work: multi-view images and point clouds. We found that algorithms using these modalities as input were difficult to fit into the scope of this work, for the following reasons:Besides CosyPose [5], we did not find any multi-view approaches that fit the scope given by our use case.Point-cloud-based object detectors are very popular in the area of autonomous driving. Consequently, they are commonly evaluated on datasets and metrics tailored to this use case (e.g., the KITTI dataset [48]), and the evaluation scores found in the literature cannot be compared to those of most RGBD-based object detectors.

However, we also found that these modalities could potentially benefit the industrial use case, particularly in these regards:Multi-view images as well as point clouds usually cover a larger portion of a scene than single-view images. Thus, they could mitigate problems due to occlusion, pose ambiguities and specular reflections.Point clouds are primarily geometrical representations of scenes, and thus, object detection based on geometrical 3D models potentially requires less preprocessing of training data, as input and training data are already in the same domain. In particular, the involved generation of synthetic images can be skipped.

For these reasons, we decided to postpone evaluating object detectors based on these modalities to future work; in particular, we plan to evaluate their performance with metrics and datasets commonly used for RGBD-based detectors.

## 4. Materials and Methods

### 4.1. Methods

In this section, we categorize the methods for 6D object detection that we have examined and compared. As this work’s scope is use case specific, we focus on method properties that put constraints on a method’s usage and regard aspects, such as the type of CNN, that were used as implementation details. In the following, we describe method properties that we found to be relevant in application scenarios and in which way they can constrain potential usages. The property description should be regarded as a set of general guidelines and not as strict rules, e.g., although depth-based detectors *tend* to give better camera–object distance estimates than RGB-based detectors, this must not be true in all circumstances. Table 1 shows our categorization of algorithms.

**Modality** describes which type of input a method accepts at training time and runtime. **RGB**-based methods tend to have a larger error when estimating the distance of objects to the camera. **Depth**-based methods are based on geometry only, so they cannot use color cues or textures visible on objects. **RGBD**-based methods can use the best of both worlds. We only regarded the modality of the core-method uses, i.e., no optional refinement steps. Of course, every RGB-based detector can be extended to RGBD by, for example, post-processing the results with ICP [47], and every depth-based detector can be extended to RGBD by employing some kind of 2D-edge-based pose refinement.**Features** states whether a method uses **learned** or **hand-crafted** features for object detection, i.e., whether the algorithm is **data-** or **model-driven**. As the name suggests, data-driven methods tend to require large amounts of training data: in our case, synthetic images. The generation of these data and the subsequent training can be computationally very demanding, in some cases needing several days for a full setup. Hand-crafted features usually do not require as much data, and the conversion of training data to features is straightforward, as no weight optimization takes place. However, the latter tend to have more parameters that need to be fine-tuned for optimal results.**Scope** describes whether a feature in the object-detection step represents the full target object (e.g., a “template”) or a single point of interest (e.g., a single pixel or an image patch). **Global** features, representing the whole object, are usually more robust when detecting multiple instances of a single object class that are close to or even occluding each other. **Local** features tend to be more robust against general occlusion or difficult lighting conditions.**Output** gives the type of space that the output pose is in. Regression-based methods predict **continuous** results, i.e., the poses they estimate are theoretically infinitely accurate. Classification-based methods predict **discrete** results, i.e., their output is one of a previously learned finite number of classes. Whether a discrete estimation is good enough depends on the use-case requirements and whether there are enough computational resources to perform a refinement step.

All methods referenced in this work have been tested with model-based training. Note that the mode of generating synthetic data differs between methods, i.e., there can be potential for better scores and thus presented scores are only lower bounds.

#### Remarks for Individual Methods

The learning-based method **FFB6D** [2] was trained on synthetic images by ourselves. To train FFB6D, we used the synthetical images generated with BlenderProc [4] for BOP Challenge 2020 [3], using scene 2 as the validation set. We deactivated all data augmentation and trained on the renderings as they are. The training ran for 366,000 iterations at a batch size of 3.**PoseRBPF** [33] is a tracking and not an object-detection method. However, the algorithm **can** actually be used for object detection (referred to as *initialization* in the respective paper), and the pose estimation accuracy is improved over consecutive frames. For this reason, we regarded it in this work, despite not fully fitting the required profile.**PointVoteNet** [30] supports both global and local features, as it is based on *PointNet* by Qi et al. [50], which represents target objects as a cascade of global and local features.The depth-based methods **PointVoteNet** [30], **PoseCluster** [22] and all **PPF**-variants [9,10,11] can be trained with point clouds only, i.e., no image synthesis is required here.

### 4.2. Datasets

In this work, we focus on datasets, which, on the one hand, pose challenges that map to the requirements stated in Section 3.2 and for which a significant amount of quantitative data are available in the literature. For all of the datasets used in this work, RGBD images with annotations for ground truth poses and 3D models of the depicted objects are available. Furthermore, synthetically generated RGBD training images are provided through BOP (see https://bop.felk.cvut.cz/home/, accessed on 30 December 2021 or [3]). Sample images of the datasets can be seen in Figure 2. These are regarded in this work:**LineMOD (LM)** **[14]:** First presented by Hinterstoisser et al. to evaluate their algorithm of the same name, the LM dataset provides 15 scenes. In each scene, 1 of 15 different objects from an office environment is annotated and placed on a desktop with severe clutter.**LineMOD occluded (LMO)** **[19]:** This dataset includes scene number 2 of the original LineMOD datasets, but with ground truth annotations for multiple objects from different classes in a single frame. In addition to the background clutter, this poses the challenge of a lot of occlusion between objects.**TLESS** **[51]:** The T-LESS dataset comprises 20 scenes with annotations for 30 different object classes. The depicted objects are all typical industrially manufactured objects, made from textureless white plastic, many of which are rotationally symmetric. The objects are all placed on a black background, so there is little background clutter. All scenes show different combinations of objects with different placements, with cases of multiple instances of one object in a scene and objects occluding each other.

For a comprehensive overview of other datasets, typically used to evaluate 6D object detection, we recommend reading the publication of Hodaň et al. [3]. Their work not only gives details on the datasets and the specific challenges they pose, but the authors also brought 12 widely used datasets in a common format and provided synthetic training images for most of them, generated using BlenderProc [4].

### 4.3. Metrics

Object detectors are usually evaluated by regarding them as binary classifiers. Consequently, metrics used to measure the performance of object detectors are calculated in two stages:The distances of detected instances and ground-truth annotations are calculated with a geometric metric. Based on a metric-specific threshold, every detected instance and ground truth annotation is classified as one of *true positive* (TP), *false positive* (FP), and *false negative* (FN).The numbers of TPs, FPs and FNs are aggregated based on a metric for the evaluation of binary classifiers, which then gives the final evaluation score.

There are several metrics commonly used in the literature to determine the performance of object detectors. We only describe metrics in detail that are relevant in this work. A metric is deemed *relevant* if it allows comparing multiple object detectors that fit the industrial use-case profile. We note that a large majority of publications considering model-based training use one of the metrics presented below. For an overview of the distribution in the literature of these metrics, consult Table 2.

The following geometric metrics are most often used in the literature to evaluate the performance of 6D object detectors that are trained on 3D models only:**Average distance (symmetric) (ADD(S))** **[14]:** This metric measures the average distance of 3D points of an object’s model transformed with two different poses. ADD-S (also *ADI*) is a variant, which takes into account that rotationally symmetric objects can have multiple valid pose estimates. ADD(S) is used to denote that the symmetric variant ADD-S is used for objects with rotational symmetries and ADD for non-symmetric objects. The most commonly used threshold for classifying an estimate as correct is t=0.1·d, where *d* is the target object’s diameter. Some publications use t=0.15, which are marked in the respective locations.**Visual surface discrepancy (VSD)** **[52]:** As the name suggests, this metric measures the difference of the *visible* surface of an object transformed with two different poses relative to the camera, i.e., if an object *looks* exactly the same when transformed with two poses, the VSD is 0. In particular, this handles rotational symmetries more intuitively than ADD(S). This metric has two threshold parameters, determining whether a pose is considered to be correct: τ is the maximum allowed difference in the camera distance of overlapping pixels; θ is the minimally allowed percentage of object pixels that need to be considered correct according to the τ condition for the whole hypothesis to be considered correct. A widely used combination of thresholds is τ=20 mm and θ=0.3. BOP Challenge 2020 [3] used a different approach by increasing τ in the interval [0.05·d,0.5·d] in steps of 0.05·d and θ in [0.05,0.5] in steps of 0.05. They then determined the score for every τ–θ pair and took the average as the total score. We refer to this configuration as **VSDBOP**.

For evaluating object detectors as binary classifiers, we found that the most widely used metrics for methods fitting our requirements profile are *recall* and *F*_1_-*score*, where the latter is the harmonic mean of *recall* and *precision*. They are calculated as follows:(2)$$recall=\frac{TPs}{TPs+FNs},\;precision=\frac{TPs}{TPs+FPs},$$
(3)$$F_{1}=2\cdot \frac{precision\cdot recall}{precision+recall}.$$

Recall is suitable for evaluating the object localization task, as defined in Section 3.1. In this case, we know that there are *N* object instances in the scene, and we have a maximum of *N* result hypotheses. From this, it follows that FPs≤FNs and recall≤precision. This makes calculating precision redundant for this task. For object detection, the number of objects to find is unknown, so here, the F1-score is required, as it takes true *and* false positives into account.

The geometric metrics that we did *not* take into account, due to their irrelevancy based on the conditions given above, but which can be found in literature include *2D projection error*, *intersection over union (IoU)*, *translational and angular error*, *maximum symmetry-aware surface distance (MSSD)*, *maximum symmetry-aware projection distance (MSPD)* and *average orientation similarity (AOS)*. The binary classification metrics that we did not regard include *average precision (AP)*, *mean average precision (mAP)* and *area under curve (AUC)*. See Hodaň et al. [3] or Sahin et al. [6] for more information on these metrics.

## 5. Evaluation

Benchmarks of object detectors are defined by three main aspects: the dataset used, the metric used and the thresholds (tolerances) used to classify a detection result as a success or failure. Searching for quantitative data on the performance of object detectors meeting the requirements we posed, we found data for the dataset–metric combinations listed in Table 2.

Although we focused on the object detection task in this work, we also examined the results for object localization benchmarks. On the one hand, there are industrial use cases for which object localization is sufficient, and on the other hand, as stated in Section 3.1, object localization can be regarded as a sub-task of 6D object detection.

Note that the empirical data we found do not address the two challenges posed in Section 3.2: unknown object color and difficult surfaces. For learning-based methods, the properties of the synthetic images used for training play a major role in the robustness against varying colors of objects, in particular, whether the renderings were generated with known colors or using some randomization strategy. We could not gather enough information on the mode of data generation for the training of all algorithms presented herein, and thus the performance regarding this aspect remains inconclusive. The same is true for reflective or translucent objects, which pose a very challenging case for all computer vision tasks and for which, to our knowledge, no annotated dataset for 6D object detection is available.

### 5.1. Discussion

In the following, we present the empirical evaluation results we found for several object detectors. We first summarize our findings regarding benchmark scores, again focusing on the requirements defined in Section 3.2, then we have a look at the runtimes of different methods, and finally take a step backward and describe our findings on the availability and comparability of empirical data in the literature.

#### 5.1.1. Method Scores

The quantitative evaluation results we found for object detectors trained on purely model-based data are reported in Table 3. From these numbers, we can draw the following conclusions, regarding the requirements posed in Section 3.2:**Object localization:** For LM-ADD(S), LM-VSD, LM-VSDBOP, TLESS-VSD and TLESS-VSDBOP, the following respective methods perform best: LCHFs [17], PPFs by Vidal et al. [11], SurfEmb [36], PoseRBPF [33] and and again SurfEmb [36]. LMO-VSDBOP allows a direct comparison of PFFs and SurfEmb, from which we can assume that the latter is the overall better method. We cannot compare the other top runners because they were not evaluated on the same metric–dataset combination, so the best overall object localizer remains inconclusive.**Object detection:** For LMO-ADD(S)-F1, LCHFs [17] perform best. As they also perform very well for object localization in LM-ADD(S), we conclude that this method can outperform many other object detectors, albeit with some reservations.**Occlusion:** The LMO-VSDBOP-ranking is led by SurfEmb [36], followed by PointVoteNet [30] and HybridPose [42] with some margin.**Workpiece-detection (textureless, rotationally symmetric):** The top runners on TLESS-VSD are PoseRBPF with SDF, followed by the same method without refinement [33], and AAE refined with ICP [43] makes third place with a large margin. For TLESS-VSDBOP, SurfEmb [36] again makes first place with a large margin, followed by HybridPose [42] and CosyPose [5].

Which is the overall best 6D pose estimator when RGBD images and only model-based training data are available? From the quantitative data, we found that we cannot answer this question. Most top performers were evaluated on different dataset–metric combinations, and thus cannot be compared based on the available data. In particular, most of the promising methods, LCHF [17], PPFs by Vidal et al. [11], SurfEmb [36] and PoseRBPF [33], cannot be compared directly to each other.

It is of specific note that under certain circumstances, hand-crafted features can still contend with learning-based methods, regardless of their age. In particular, for LM-ADD(S), the ten-year-old LineMOD algorithm [14] and its variant by Rios-Cabrera [15] very nearly reach the performance of LCHFs [17] and outperform a lot of other newer methods (e.g., SSD6D [44], SynPo-Net [32] and AAEs [43]). For LM-VSD, the PPF-based method by Vidal et al. [11] still outperforms all other methods, and for TLESS-VSD, it makes second place. However, the good performance in this ranking needs to be put under some reservations:A lot of newer methods in the literature are trained on real or a combination of real and synthetic data, and for a lot of generally promising methods, there are currently no or little empirical data available on the performance with purely model-based training; if data are available, they are not comparable.LineMOD as well as PPFs have drawbacks compared to learning-based methods that are not reflected in the scores, such as the need for manual parameter optimization (both), fragility against occlusion (LineMOD) and slow runtimes (PPF).Both LineMOD and PPFs show mediocre performance for LM-F1, while being good at generating high recalls. We assume this is because both methods are not discriminative (i.e., they do not explicitly “know” what to *exclude*), and thus tend to have lower precision than learning-based methods.

#### 5.1.2. Runtime

In Table 4, we list the runtimes for the evaluated methods, if they were available. The top two performers and the only ones that reach a frame rate for interactive real-time applications are methods based on neural networks that work on RGB images without any refinement [24,32], which comes as no surprise. The fastest RGBD-based method is the LineMOD-variant DTT-OPT-3D [15]. The numbers show that ICP-refinement is a costly operation. Methods relying on ICP generally perform worse and more specifically, we can see a difference of 0.6 s when we compare AAE, with and without refinement. The slowest methods are those based on PPFs. It is of specific note that SurfEmb, the top-performer regarding occlusion and workpiece detection, needs about 9 seconds per frame, which, for many use cases, is not acceptable.

#### 5.1.3. Availability and Comparability of Empirical Data

In Table 2, we list the amount of data we have for each dataset–metric combination. We observe a focus on recall-based metrics in literature. Recall-based metrics have a total of 70 rows of data, while the $F_{1}$-score, which also considers precision, only has 7, i.e., many publications only evaluate the performance of their algorithms regarding object localization, ignoring false positives, which can be a significant problem in object detection scenarios.

Many methods cannot be compared due to being evaluated on different datasets or metrics. We assume that one major reason for this is the effort required for processing multiple datasets or implementing different metrics. It would be desirable to have a benchmarking framework with a well-defined interface for datasets and pose-estimation results, supporting multiple metrics and allowing simple extension thereof. This framework should be accompanied by an online database that allows easy collection and analysis of empirical data on object detection performance. BOP is a big step in this direction, especially regarding the standardization of datasets. However, the evaluation metrics implemented herein were novel at the time of publication, and so they are not comparable to older results. Additionally, the BOP challenges do not regard precision in their benchmarks. In Appendix A.1 of [3], the authors discussed their decision on excluding precision in their benchmark and came to the conclusion that, for the purpose of their benchmark, recall-based scores are appropriate for two reasons. First, these scores were not saturated at the time of writing and second, only regarding the recall is less computationally complex for the evaluation framework. We consider these valid arguments for a benchmark targeting the research community, but argue that for the potential *application* of object detectors, more information is needed in order to determine its suitability when considering the requirements for a specific use case.

## 6. Conclusions and Future Work

In this work, we surveyed the state of the art of 6D object detection with a focus on industrial applications for which we identified model-based training and support for RGBD images as being especially important. We presented a collection of qualitative and quantitative information on object detectors from the literature and new data for the object detector FFB6D [2]. These data were discussed, and promising candidates for taking on specific challenges of industrial applications were identified.

Furthermore, we had a look on the availability of data that provide information on the suitability of algorithms for the use case we analyzed. We found that, for many methods, there are not enough empirical data available in the literature to determine how well they are suited to tackle specific challenges or to compare them to competing methods. In particular, many methods in the literature were only tested on real data, and many methods were only evaluated on recall-based metrics.

Based on these findings, we believe that the following future research topics would benefit the application of 6D object detection in industrial contexts:Train established and promising object detectors with model-based data and evaluate them.Evaluate established and promising object detectors with metrics that take precision into consideration.Take methods based on point clouds and multi-view images into consideration.Allow researchers to produce meaningful and comparable data by providing tools and frameworks that offer uniform formats and interfaces for evaluating object detectors on a multitude of different datasets and metrics. Additionally, provide an online database that simplifies collecting, categorizing and analyzing evaluation results. We consider BOP to be a good start in this direction, but in order to be a general-purpose framework for evaluating object detection, it should be extended with more metrics and simpler interfaces.

## Figures and Tables

**Figure 1 jimaging-08-00053-f001:**
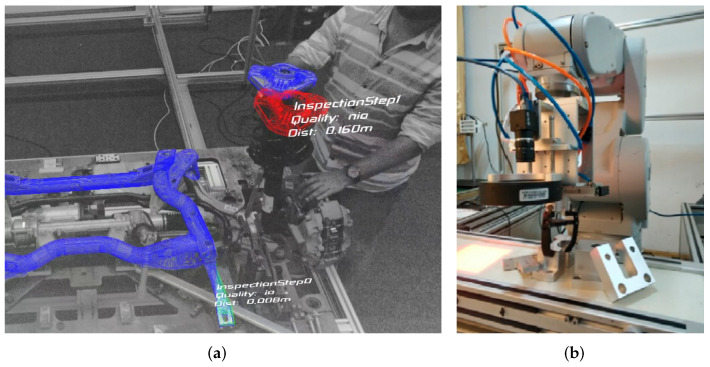
Two examples for applying object detectors in industrial production processes. (**a**) Quality control. (**b**) Bin picking (image from [1] published under CC BY 4.0).

**Figure 2 jimaging-08-00053-f002:**
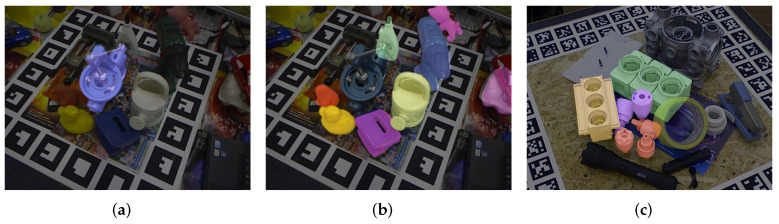
Sample images from the three datasets considered in this work. Lighter renderings show annotated poses. (**a**) LM, (**b**) LMO, (**c**) TLESS. Images used with permission from Hodaň et al. [3].

**Table 1 jimaging-08-00053-t001:** Properties of 6D object detectors. Methods are sorted alphabetically. For detailed explanations of properties, see Section 4.1. Alternating grey and white rows are visual aids.

Method	By	Year	Modality	Features	Scope	Output
AAE	Sundermeyer et al. [43]	2020	RGB	Learned	Global	Cont.
CAE	Kehl et al. [21]	2016	RGBD	Learned	Local	Cont.
CDPNv2	Li et al. [29]	2019	RGBD	Learned	Local	Cont.
CosyPose	Labbé et al. [5]	2020	RGB	Learned	Global	Cont.
DPOD	Zakharov et al. [27]	2019	RGB	Learned	Local	Cont.
DTT-OPT-3D	Rios-Cabrera and Tuytelaars [15]	2013	RGBD	Learned	Global	Disc.
EPOS	Hodaň et al. [3]	2020	RGB	Learned	Local	Cont.
FFB6D	He et al. [2]	2021	RGBD	Learned	Local	Cont.
HybridPose	König and Drost [42]	2020	RGBD	Learned	Global	Cont.
LCHF	Tejani et al. [16]	2014	RGBD	Learned	Local	Cont.
LCHF	Tejani et al. [17]	2018	RGBD	Learned	Local	Cont.
LineMOD	Hinterstoisser et al. [14]	2013	RGBD	Hand-crafted	Global	Disc.
ObjPoseFromSyn	Rambach et al. [23]	2018	RGB	Learned	Global	Cont.
Pix2Pose	Park et al. [26]	2019	RGB	Learned	Local	Cont.
PointVoteNet	Hagelskjar and Buch [30]	2020	D	Learned	Both	Cont.
PoseCluster	Buch et al. [22]	2017	D	Learned	Local	Cont.
PoseRBPF	Deng et al. [33]	2021	RGBD	Learned	Global	Cont.
PPF	Drost et al. [9]	2010	D	Hand-crafted	Local	Cont.
PPF	Hinterstoisser et al. [10]	2016	D	Hand-crafted	Local	Cont.
PPF	Vidal et al. [11]	2018	D	Hand-crafted	Local	Cont.
PVNet	Peng et al. [49]	2019	RGB	Learned	Local	Cont.
RandomForest	Brachmann et al. [19]	2014	RGB	Learned	Local	Cont.
SSD6D	Kehl et al. [44]	2017	RGB	Learned	Global	Cont.
SurfEmb	Haugaard and Buch [36]	2021	RGBD	Learned	Global	Cont.
SyDPose	Thalhammer et al. [28]	2019	D	Learned	Global	Cont.
SynPo-Net	Su et al. [32]	2021	RGB	Learned	Global	Cont.
TemplateBased	Hodan et al. [18]	2015	RGBD	Hand-crafted	Global	Cont.
UncertaintyDriven	Brachmann et al. [20]	2016	RGB	Learned	Local	Cont.
YOLO6D	Tekin et al. [24]	2018	RGB	Learned	Local	Cont.

**Table 2 jimaging-08-00053-t002:** Dataset–metric combinations we found empirical data for, the tasks (Section 3.1) and challenges (Section 3.2) that they address, and the number of data points we found for each (i.e., the number relevant methods evaluated). Alternating grey and white rows are visual aids.

Dataset	Metric	Task	Challenges	Data-Points
LM	ADD(S)-Recall	Localization	Background-clutter	19
LM	ADD(S)-F1	Detection	Background-clutter	7
LM	VSD-Recall	Localization	Background-clutter	11
LMO	VSDBOP-Recall	Localization	Background-clutter, occlusion	17
TLESS	VSD-Recall	Localization	Texturelessness, symmetry	11
TLESS	VSDBOP-Recall	Localization	Texturelessness, symmetry	12

**Table 3 jimaging-08-00053-t003:** Scores of model-based trained 6D object detectors for different datasets and metrics in percentages. Methods are sorted alphabetically for easy cross referencing with Table 1. Methods variants are given in brackets (e.g. refinements, like ICP). Unlike in Table 1, refinement steps are considered in the modality-column. All except ADD(S)-F1 are recall-based scores. Darker green in cell backgrounds denotes higher scores, column wise; alternating grey and white rows are visual aids. The top three methods of a column are in **bold**. † denotes that a threshold of t=0.15·d was used for ADD(S) instead of t=0.1·d. No citation is given for the values of FFB6D, as we evaluated them ourselves.

		LM	LM	LM	LMO	TLESS	TLESS
Method	Modality	ADD(S)	ADD(S)-F1	VSD	VSDBOP	VSD	VSDBOP
**AAE** **[43]** (ICP)	RGBD	71.58 [32]				**69.53** [43]	
**AAE [43]**	RGB	32.63 [32]				20.53 [43]	
**CAE [21]** (ICP)	RGBD			58.2 [12]		24.6 [12]	
**CDPNv2 [29]** (ICP)	RGBD				46.9 [53]		36.8 [53]
**CDPNv2 [29]**	RGBD				44.5 [53]		30.3 [53]
**CosyPose [5]**	RGB				48 [53]		**57.1** [53]
**DPOD [27]**	RGB				10.1 [53]		4.8 [53]
**DTT-OPT-3D [15]**	RGBD	**96.5** [6]					
**EPOS [3]**	RGB				38.9 [53]		38 [53]
**FFB6D [2]**	RGBD	54.08	55.5		37.7		
**HybridPose [42]** (ICP)	RGBD				**51.7** [53]		**58** [53]
**LCHF [16]** (co-tra)	RGBD	78.6 [6]	**82** [6]				
**LCHF [16]**	RGBD			12.1 [12]			
**LCHF [17]** (Iterated)	RGBD		**81.7** † [17]				
**LCHF [17]**	RGBD	**98.2** † [17]	**78.8** † [17]				
**LineMOD [14]** (ICP)	RGBD	96.3 [6]					
**LineMOD [14]**	RGBD	**96.6** [14]	63 [6]				
**ObjPoseFromSyn [23]**	RGB	10.22 [32]					
**Pix2Pose [26]**	RGB	11.32 [32]			15.6 [53]		
**PointVoteNet [30]** (ICP)	D				**53.5** [53]		0.3 [53]
**PoseCluster [22]** (ICP, PPFH)	D			56.6 [12]			
**PoseCluster [22]** (ICP, SI)	D			33.33 [12]			
**PoseRBPF [33]** (SDF)	RGBD					**82.58** [33]	
**PoseRBPF [33]**	RGBD					**80.52** [33]	
**PPF [9]** (Edge)	RGBD				42.5 [53]	67.5 [43]	46.9 [53]
**PPF [9]** (ICP)	D			**82** [12]	43.7 [53]		37.5 [53]
**PPF [9]** (ICP, Edge)	RGBD			79.13 [12]	39.2 [53]		37 [53]
**PPF [9]**	D	78.9 [6]	51.7 † [17]			56.81 [43]	
**PPF [10]**	D	96.4 [6]					
**PPF [11]** (ICP)	D			**87.83** [12]	47.3 [53]	66.51 [43]	46.4 [53]
**PPF [11]**	D					66.3 [33]	
**PVNet [49]** (ICP)	RGBD				50.2 [53]		
**PVNet [49]**	RGB				42.8 [53]		
**RandomForest [19]**	RGB			67.6 [12]			
**SSD6D [44]** (ICP)	RGBD	79 [32]					
**SSD6D [44]**	RGB	2.42 [32]			4.7 [53]		
**SurfEmb [36]**	RGBD				**61.5** [53]		**79.7** [53]
**SyDPose [28]**	D	30.21 [28]	59.1 † [28]				
**SynPo-Net [32]** (ICP)	RGBD	72.29 [32]					
**SynPo-Net [32]**	RGB	44.13 [32]					
**TemplateBased [18]** (PSO)	RGBD	94.9 [6]		**87.1** [12]			
**TemplateBased [18]**	RGBD			69.83 [12]		63.18 [43]	
**UncertaintyDriven [20]**	RGB			75.33 [12]		17.84 [43]	
**YOLO6D [24]**	RGB	21.43 [32]					

**Table 4 jimaging-08-00053-t004:** Runtimes of methods and their variants. Note that the comparability of the values listed here is under great reservations, as they were generated by different persons with different parameters, using different hardware over the course of 11 years. Nevertheless, they can at least give an impression of the order of magnitude in which algorithms perform. Alternating grey and white row colors are visual aids.

Method	Variant	Modality	Runtime [s]
SynPo-Net [32]		RGB	0.015
YOLO6D [24]		RGB	0.02
DTT-OPT-3D [15]		RGBD	0.055
PoseRBPF [33]		RGBD	0.071
SSD6D [44]		RGB	0.083
PoseRBPF [33]	SDF	RGBD	0.156
FFB6D [2]		RGBD	0.196
AAE [43]		RGB	0.2
DPOD [27]		RGB	0.206
HybridPose [42]	ICP	RGBD	0.337
CosyPose [5]	RGB	RGB	0.47
CosyPose [5]		RGB	0.493
AAE [43]	ICP	RGBD	0.8
CDPNv2 [29]		RGB	0.98
PPF [9]	ICP	D	1.38
LCHF [16]		RGBD	1.4
RandomForest [19]		RGBD	1.4
CDPNv2 [29]	ICP	RGBD	1.49
CAE [21]	ICP	RGBD	1.8
EPOS [3]		RGB	1.87
LCHF [17]		RGBD	1.96
TemplateBased [18]	PSO	RGBD	2.1
PPF [9]		D	2.3
PPF [11]	ICP	D	3.22
UncertaintyDriven [20]		RGBD	4.4
SurfEmb [36]		RGBD	9.227
TemplateBased [18]		RGBD	12.3
PoseCluster [22]	ICP, PPFH	D	14.2
PoseCluster [22]	ICP, SI	D	15.9
PPF [9]	Edge	RGBD	21.5
PPF [9]	ICP, Edge	D	21.5
PPF [9]	ICP	RGBD	87.57

## Data Availability

Not applicable.

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
