# Peer review of "A Survey of 6D Object Detection Based on 3D Models for Industrial Applications"

_2313-433X, 2022, doi:10.3390/jimaging8030053_

Round 1

Reviewer 1 Report

The paper is a literature review on 6d pose estimation specifically for industrial objects. The authors provided a listing of requirements typical for such applications. They also listed the related datasets and evaluation metrics that have been used in the literature.

6d pose estimation in industrial settings is getting more attentions in recent years and can be challenging due to occlusion, object symmetry and having less visual cues. 

Although the authors looked at the problem from different perspective, I think the literature review is not comprehensive, more specifically the papers that have been working on Tless dataset, which is designed for industrial part applications. The authors did not discuss on point cloud based approaches as well. Comparison on available metrics is also missing.

Author Response

Hello and many thanks for your review. We revised the paper and addressed your review in the following way. You can find the revised version in the attachment.

Point 1: Although the authors looked at the problem from different perspective, I think the literature review is not comprehensive,

We agree that this is not a comprehensive review and think, depending on the definition, “Survey” would be more appropriate. We decided to use “Review”, because this was the best fit from MDPI’s article types. We changed the title to “A Survey of 6D Object Detection based on 3D-Models for Industrial Applications” and adapted respective references in text. 

Point 2: … more specifically the papers that have been working on Tless dataset, which is designed for industrial part applications. 

We included all methods we found, that fit our requirements-profile (mainly: tested with model-based training).

Point 3: The authors did not discuss on point cloud based approaches as well. 

We added section 3.4, describing modalities in more detail. Here we also explain why we did not include multi-view and point-cloud based methods. Note, as the D-channel of RGBD can easily be converted to a point-cloud, we regarded PointVoteNet, PoseClustering and PPF-variants, which could also be used on full-scene-point-clouds.

Point 4: Comparison on available metrics is also missing. 

We extended the section on metrics (4.3) to explicitly explain metrics for binary classification and explain which metrics we did not consider in more detail at the end of the section.

List of all major changes, including those based on other reviews:

  • General: Changed wording: "review" to "survey" in title and when referencing paper.
  • General: Fixed grammar, spelling and phrasing errors.
  • 1: Explained the position of this work in contrast to prior reviews and benchmarks.
  • 1: Provided a definition for "6D object detection" also explaining the relationship to "pose estimation" and "object localization". Consequently replaced most usages of "pose estimation" with "object detection".
  • 2.: Removed "single-instance-" vs. "multi-instance-detection" in favor of "object localization" vs. "object detection" and adapted the rest of the paper to the slightly different semantics.
  • 3: Revised section on "synthetic training" to "model-based training". Differentiate better by replacing most usages of "synthetic training" with "model-based training" and "synthetic data" with "3D-models". Only use "synthetic data" to refer to synthetically generated images.
  • 4: Added "Modalities"-section, including rationale for regarding RGBD- and RGB-based methods.
  • 4: Added rationale for not including a lot of point-cloud-based and multi-view-based methods.
  • 1: Added a description of different method categories and categorized methods accordingly.
  • 3: Wrote a more detailed overview over metrics. Explicitly explain metrics used to evaluate binary classifiers. Explicitly list metrics that were not considered and explain why.
  • 5: Replaced listing of dataset-metric-combinations with table, that gives a better overview.
  • 1.1: Made all rows of scoring-table the same height and added more explanation in caption.
  • 1.3: Moved number of data-points per dataset-metric combination to table in 5.
  • References: fixed wrong classifications as "technical reports", cited conference-proceedings instead of lecture notes.

Reviewer 2 Report

This paper presents interesting and valuable knowledge on the 6D pose estimation topic, but several points make it unclear.

In Related Works, the position of this paper is not well-clarified, compared to other in-depth review papers, which are not listed in this paper.

As a major flaw, from Section 2.2 to Section 5, all estimators considered in this paper are not well-categorized. This work basically assumes RGBD- and learning-based methods for 6D pose estimation, but handcraft feature-based methods are mixed up together (do handcraft ones need synthetic data?). RGB-based methods are also listed and compared.

Similarly, (As mentioned in Section 5.1.1) in this work, it seems that the detection (or recognition) task is considered as a part of a pose estimation task, but actually, it is not (as you know). In this regard, all datasets/metrics considered in this paper can actually be more proper for detection-pose estimation schemes than "Actual" pose estimation schemes that rely on temporal knowledge between interframes. In some cases, the detection performance can more be contributed to overall performance; thus, I definitely understand that its clear separation is not necessary in practice, but such an aspect is not clarified.

Minors:
- I am not sure that we can call this paper "a review paper" because this paper does not include comprehensive explanations (including pros and cons) about the current state of methods on this topic.

- The visibility of Table 1 is not good. Methods look uncategorized, including repeated naming. Also, what do Gery darker cell-backgrounds mean?

- In Reference, there are some errors, e.g., [6] is not a technical report. Also, please use the conference name instead of the Lecture Notes series.

- English should be carefully proofread again.

Author Response

Hello and many thanks for your detailed review, it was very helpful and constructive. We revised the paper and addressed your points with the following changes. You can find the revised version in the attachment.

Point 1: In Related Works, the position of this paper is not well-clarified, compared to other in-depth review papers, which are not listed in this paper. 

We added a paragraph addressing this point in section 2.1.

Point 2: As a major flaw, from Section 2.2 to Section 5, all estimators considered in this paper are not well-categorized. This work basically assumes RGBD- and learning-based methods for 6D pose estimation, but handcraft feature-based methods are mixed up together (do handcraft ones need synthetic data?). RGB-based methods are also listed and compared. 

We added a categorization of methods and the effect of different categories on our use-case in section 4.1. We also added section 3.4 on modalities, explaining our selection of modalities.

Point 3: … do handcraft ones need synthetic data? 

We revised our definition of “synthetic data”/”synthetic training”, most importantly replacing it with “3D-models”/”model-based training” where appropriate. You can find the definition in section 3.2.

Point 4: Similarly, (As mentioned in Section 5.1.1) in this work, it seems that the detection (or recognition) task is considered as a part of a pose estimation task, but actually, it is not (as you know). In this regard, all datasets/metrics considered in this paper can actually be more proper for detection-pose estimation schemes than "Actual" pose estimation schemes that rely on temporal knowledge between interframes. In some cases, the detection performance can more be contributed to overall performance; thus, I definitely understand that its clear separation is not necessary in practice, but such an aspect is not clarified. 

We added a more detailed definition of “object detection”, “pose estimation” and “object localization” and their relationships in section 3.1 and revised the usage of these terms accordingly.

Point 5: - I am not sure that we can call this paper "a review paper" because this paper does not include comprehensive explanations (including pros and cons) about the current state of methods on this topic. 

I agree that this is not a comprehensive review and think, depending on the definition, “Survey” would be more appropriate. I decided to use “Review”, because this was the best fit from MDPI’s article types. However, I will change the title to “A Survey of 6D Object Detection based on 3D-Models for Industrial Applications”. 

Point 6: - The visibility of Table 1 is not good. Methods look uncategorized, including repeated naming. Also, what do Gery darker cell-backgrounds mean? 

Made all rows of the scoring-table the same height for a less noisy appearance and added explanations on design-decision in the caption.

Point 7: - In Reference, there are some errors, e.g., [6] is not a technical report. Also, please use the conference name instead of the Lecture Notes series. 

Done.

Point 8: - English should be carefully proofread again. 

Also Done.

List of all major changes, including those based on other reviews:

  • General: Changed wording: "review" to "survey" in title and when referencing paper.
  • General: Fixed grammar, spelling and phrasing errors.
  • 1: Explained the position of this work in contrast to prior reviews and benchmarks.
  • 1: Provided a definition for "6D object detection" also explaining the relationship to "pose estimation" and "object localization". Consequently replaced most usages of "pose estimation" with "object detection".
  • 2.: Removed "single-instance-" vs. "multi-instance-detection" in favor of "object localization" vs. "object detection" and adapted the rest of the paper to the slightly different semantics.
  • 3: Revised section on "synthetic training" to "model-based training". Differentiate better by replacing most usages of "synthetic training" with "model-based training" and "synthetic data" with "3D-models". Only use "synthetic data" to refer to synthetically generated images.
  • 4: Added "Modalities"-section, including rationale for regarding RGBD- and RGB-based methods.
  • 4: Added rationale for not including a lot of point-cloud-based and multi-view-based methods.
  • 1: Added a description of different method categories and categorized methods accordingly.
  • 3: Wrote a more detailed overview over metrics. Explicitly explain metrics used to evaluate binary classifiers. Explicitly list metrics that were not considered and explain why.
  • 5: Replaced listing of dataset-metric-combinations with table, that gives a better overview.
  • 1.1: Made all rows of scoring-table the same height and added more explanation in caption.
  • 1.3: Moved number of data-points per dataset-metric combination to table in 5.
  • References: fixed wrong classifications as "technical reports", cited conference-proceedings instead of lecture notes.

Reviewer 3 Report

The paper presents a review of the state of the art of 6D pose estimation with a focus on industrial applications that utilize RGB D cameras. In my opinion, the review is extensive and well prepared. I have no comments on the article.

Author Response

Hello and many thanks for your positive review. You can find a revised version of the paper in the attachments. These are the major changes based on other reviews:

  • General: Changed wording: "review" to "survey" in title and when referencing paper.
  • General: Fixed grammar, spelling and phrasing errors.
  • 1: Explained the position of this work in contrast to prior reviews and benchmarks.
  • 1: Provided a definition for "6D object detection" also explaining the relationship to "pose estimation" and "object localization". Consequently replaced most usages of "pose estimation" with "object detection".
  • 2.: Removed "single-instance-" vs. "multi-instance-detection" in favor of "object localization" vs. "object detection" and adapted the rest of the paper to the slightly different semantics.
  • 3: Revised section on "synthetic training" to "model-based training". Differentiate better by replacing most usages of "synthetic training" with "model-based training" and "synthetic data" with "3D-models". Only use "synthetic data" to refer to synthetically generated images.
  • 4: Added "Modalities"-section, including rationale for regarding RGBD- and RGB-based methods.
  • 4: Added rationale for not including a lot of point-cloud-based and multi-view-based methods.
  • 1: Added a description of different method categories and categorized methods accordingly.
  • 3: Wrote a more detailed overview over metrics. Explicitly explain metrics used to evaluate binary classifiers. Explicitly list metrics that were not considered and explain why.
  • 5: Replaced listing of dataset-metric-combinations with table, that gives a better overview.
  • 1.1: Made all rows of scoring-table the same height and added more explanation in caption.
  • 1.3: Moved number of data-points per dataset-metric combination to table in 5.
  • References: fixed wrong classifications as "technical reports", cited conference-proceedings instead of lecture notes.

Round 2

Reviewer 1 Report

Thank you for modifying your work. It is a much better read now. I still believe that a comprehensive survey paper that consider different approaches (for the same problem and application) would be more beneficial and informative to the reader. However, the current manuscript can provide insights for the readers and therefore I recommend its publications.

Author Response

Hello and thanks for the re-review! We plan to do a more comprehensive review in future work, providing a broader view of the research field. We also plan on performing more experiments ourselves, to fill the gaps we found and to get better comparability by using a common training-set.

Reviewer 2 Report

This revision looks understandable with more clarification and explanation. For the sake of the main interest and scope in this paper, technical terms are more clearly defined, and the categorization of methods is improved.

In this study, it might be the weakness that the findings are inconclusive and limited based on a collection of performance results, ​but I think this paper could provide informative insights and valuable discussion when considering this issue in not only research communities but also practical fields.

Author Response

Hello and thanks for the re-review. We'll do a more comprehensive benchmark in future work, performing experiments ourselves, to fill the gaps we found. This would also allow us to ensure better comparability, by using a common training-set.